# Total Eclipse of the Zoo: Animal Behavior during a Total Solar Eclipse

**DOI:** 10.3390/ani10040587

**Published:** 2020-03-31

**Authors:** Adam Hartstone-Rose, Edwin Dickinson, Lisa M. Paciulli, Ashley R. Deutsch, Leon Tran, Grace Jones, Kaitlyn C. Leonard

**Affiliations:** 1Department of Biological Sciences, North Carolina State University, Raleigh, NC 27695, USA; ekdickin@ncsu.edu (E.D.); lmpaciul@ncsu.edu (L.M.P.); kcleona2@ncsu.edu (K.C.L.); 2Department of Anthropology, University of Florida, Gainesville, FL 32611, USA; adeutsch@ufl.edu; 3School of Life Sciences, University of Hawaii at Manoa, Honolulu, HI 96822, USA; leontran@hawaii.edu; 4School of Medicine, University of South Carolina, Columbia, SC 29209, USA; grace.jones@uscmed.sc.edu

**Keywords:** weather, zoo, anxiety, light, captive

## Abstract

**Simple Summary:**

A total solar eclipse is a rare but impactful meteorological event that has been historically associated with anomalous behavioral responses within animals. In this study, we compare the behaviors of 17 species (including mammals, birds, and reptiles) during a total solar eclipse at the Riverbanks Zoo in Columbia, South Carolina, USA to baseline behavioral observations collected prior to the event. Behavioral responses were classified into one or more of four categories: normal/baseline, evening, novel, and apparent anxiety. Approximately 75% of observed species exhibited a behavioral response to the eclipse, with the majority of these animals engaging in their established evening or nighttime behaviors. The next most frequent response was apparent anxiety. These observations provide new data on the behavioral impact of this meteorological phenomenon across a diverse group of animals, which may prove useful in contextualizing future observations.

**Abstract:**

The infrequency of a total solar eclipse renders the event novel to those animals that experience its effects and, consequently, may induce anomalous behavioral responses. However, historical information on the responses of animals to eclipses is scant and often conflicting. In this study, we qualitatively document the responses of 17 vertebrate taxa (including mammals, birds, and reptiles) to the 2017 total solar eclipse as it passed over Riverbanks Zoo and Garden in Columbia, South Carolina. In the days leading up to the eclipse, several focal teams, each consisting of researchers, animal keepers, and student/zoo volunteers conducted baseline observations using a combination of continuous ad libitum and scan sampling of each animal during closely matched seasonal conditions. These same focal teams used the same protocol to observe the animals in the hours preceding, during, and immediately following the eclipse. Additionally, for one species—siamangs (*Symphalangus syndactylus*)—live video/audio capture was also employed throughout observations to capture behavior during vocalizations for subsequent quantitative analysis. Behavioral responses were classified into one or more of four overarching behavioral categories: normal (baseline), evening, apparent anxiety, and novel. Thirteen of seventeen observed taxa exhibited behaviors during the eclipse that differed from all other observation times, with the majority (8) of these animals engaging in behaviors associated with their evening or nighttime routines. The second predominant behavior was apparent anxiety, documented in five genera: baboons (*Papio hamadryas*), gorillas (*Gorilla gorilla gorilla*), giraffes (*Giraffa cf. camelopardalis)*, flamingos (*Phoenicopterus ruber*), and lorikeets (*Trichoglossus moluccanus* and *Trichoglossus haematodus*). Novel behaviors characterized by an increase in otherwise nearly sedentary activity were observed only in the reptiles, the Galapagos tortoise (*Chelonoidis nigra*) and the Komodo dragon (*Varanus komodoensis*). While the anthropogenic influences on animal behaviors—particularly those relating to anxiety—cannot be discounted, these observations provide novel insight into the observed responses of a diverse vertebrate sample during a unique meteorological stimulus, insights that supplement the rare observations of behavior during this phenomenon for contextualizing future studies.

## 1. Introduction

The astronomical phenomenon of a total solar eclipse has long been of interest to animal behaviorists (e.g., [1,2,3,4,5,6,7,8]) and indeed to the broader public and scientific community. Due to the rare nature of this event—a total eclipse of the sun by the moon occurs in the same place only once in a ~375 year timespan [9]—this phenomenon is novel to those animals that experience its effects and, consequently, may induce anomalous behaviors. Historically, information regarding behavioral changes related to eclipses has been largely anecdotal. Nonetheless, several such accounts provide a valuable insight into the behavioral shifts associated with this occurrence. 

The most comprehensive historical accounts of animal behavior during such an event are derived from observations of animals during the total solar eclipse of New England in 1932 [1]. During this event, many animals were described as participating in “nighttime” behaviors, such as returning to their nests and hives, the commencement of evening vocalization patterns, or (in crepuscular or nocturnal taxa) an increase in activity patterns. Reactive behaviors associated with the onset of night were similarly observed in both domesticated (e.g., sheep and cattle) and wild (e.g., bats, moths, and birds) species. These data strongly suggest the ability of this event—despite its brief nature—to disrupt the circadian rhythms of a broad diversity of animals, from insects to birds and mammals. 

Similar behaviors have been observed in both wild and domestic animals in more recent eclipse events. During a solar eclipse in Bharatpur, India, in 1995, night herons were observed to abandon their daylight roosts [10], while, in Venezuela in 1998, diurnal birds such as pelicans sought out their evening roosts during totality [11]. Similarly, during an eclipse over Kansas in 1994, at least four species of diurnal birds (great egrets, cattle egrets, snowy egrets, and little blue herons) were reported to instigate their evening routines at the onset of the event [12].

The onset of evening/night behaviors, however, is documented to be far from a universal response; several taxa have been reported to exhibit behaviors normally associated with anxiety during the duration of an eclipse, suggesting a fearful or otherwise negative response to the event. Such reactions were observed in domestic dogs (who were recorded to fall silent during the eclipse) [1] and horses (who clustered, and began shaking their heads and tails), as well as in several species of wild birds (crows, gulls, and sparrows) who ceased flying and remained silent and still [13]. More unorthodox behaviors were observed among primates: in India, a large group of rhesus macaques was observed to splinter into smaller subgroups and settle down to sleep until reuniting at the conclusion of the eclipse [14], while a captive group of chimpanzees were reported to climb the highest structures in their enclosure, orienting themselves toward the sky throughout the duration of totality in Georgia during the 1984 eclipse [15].

In contrast, a number of reports indicate a more muted (and often non-registered) response to an eclipse. A broad analysis of animal behavior during a 1955 eclipse of India asserted that neither wild nor captive animals exhibited a meaningful response to the event [16]. While baseline behaviors were not reported for these animals (making such an assertion difficult to evaluate), several additional accounts of animal behavior during an eclipse event relayed similar findings. Four out of five rodents analyzed during a 1980 eclipse in India exhibited no discernible response to the event; the fifth, the Indian bush rat, exhibited responses attributed to anxiety [17]. Similarly, a diverse range of animals observed during the 2001 eclipse in Zimbabwe (including warthogs, crocodiles, zebra, eland, and lions) were reported to demonstrate no discernible behavioral response to the eclipse [18].

Collectively, these observational data suggest that the behavioral responses of animals during an eclipse may be complex, confusing, and often contradictory, as in the case of black-crowned night herons, which have been documented in one report to have responded noisily to the eclipse [10] but in another to have exhibited no response [11]. Nonetheless, these observations indicate that animals may react to an eclipse phenomenon in one of four overarching ways: (1) by commencing evening routines/exhibiting nighttime behaviors; (2) by exhibiting responses commensurate with fear and/or anxiety; (3) by exhibiting a novel behavioral response; (4) by exhibiting no discernible reaction to the environmental change.

In this study, we use this broad paradigm of classification to investigate the responses of seventeen species of mammals, birds, and reptiles housed at The Riverbanks Zoo and Garden in Columbia, South Carolina, to the total solar eclipse of 21 August 2017. This event, which traveled across the entire continental United States from Oregon to South Carolina, captured the zoo within the path of totality, resulting in a total solar eclipse that produced approximately 2.5 min of near-total darkness. Taxa were selected for observation using a combined criteria including (1) taxa for whom previous observations were made during eclipses, (2) animals that were known to be sensitive to day-cycle and weather phenomena, and (3) taxa with observable dramatic behavior shifts during normal activity (see methods).

Using baseline behaviors derived from earlier first-hand observations of the same animals in their enclosures and from detailed discussions with their primary keepers, we describe the behaviors exhibited by each animal during the astrological event relative to their typical routines, and classify these behaviors into one or more of the four broad response scenarios outlined previously. While the short and non-repeatable nature of the eclipse renders quantitative behavioral comparisons largely impossible, we utilize standardized techniques where possible (e.g., when measuring call frequencies and durations) to maximize consistency and scientific rigor. However, for many animals, our behavioral data are restricted to qualitative, observational accounts of their actions (or inaction).

## 2. Materials and Methods

### 2.1. Sample

Animal observations were conducted by several focal teams, each consisting of one or more university researchers, advised by a member of zoo staff and supplemented with zoo volunteers and university students. In total, these focal teams comprised 21 university affiliates, 19 zoo staff members, and 13 zoo volunteers, with each focal group containing at least one representative of each category. Prior to the commencement of observations, all members of a focal team were trained in observational methodologies and study procedures to standardize reports between teams. This training involved familiarization with the animals and their enclosures, their typical behavioral repertoire, and the methods of recording each animal’s movements and activities during the observation period (detailed below). However, tests of interobserver reliability for behavioral assessments were not conducted. Following training, each focal team was assigned one animal exhibit, and maintained observation of that single enclosure throughout the course of the entire study. Following the criteria outlined above, a total of 17 focal taxa were observed: among mammals, three primate species (hamadryas baboons, siamangs, and gorillas), African elephants, grizzly bears, giraffes, harbor seals and California sea lions; among birds, American flamingos, lorikeets, cockatoos, lapwings, kookaburras, and tawny frogmouths; and among reptiles, Galapagos tortoises and Komodo dragons. All animals observed were adults with the exception of the large lorikeet and flamingo flocks—each of which included fledged juveniles. All animals were housed in fully outdoor enclosures, and thus experienced the effects of the eclipse directly. 

### 2.2. Observations

Animal observations began two days prior to the eclipse, to enable baseline data to be collected that would directly mirror both the seasonal and climatic conditions of the eclipse, as well as ensure that group composition and exhibit layout remained consistent throughout all observations. Moreover, the two days prior to the eclipse—being in the middle of summer—encompassed the zoo’s highest-attendance weekend, which simulated the unusually high volume of visitors at the zoo during the day of the eclipse. On each day (the two days prior to the eclipse, and the day of the eclipse itself), animals were observed for three hours (13:00 to 16:00), which bracketed the time of the eclipse’s totality on the final day.

During observation periods, a focal animal ad libitum activity was recorded for most taxa and group activity was recorded for the two bird exhibits (lorikeets and flamingos) for which the flocks were too large to comprehensibly localize behaviors to specific individuals. Data were transcribed in field notebooks throughout the observation period. The observations employed continuous sampling methodologies supplemented by periodic scan sampling at thirty-minute intervals to make sure that data were updated for all members in each group [19,20]. Changes in the location and/or behavior of an animal were recorded. In addition, notes on each animal’s status were made at thirty-minute intervals throughout observations. For one species (the siamangs), live video and audio recording was employed throughout the observational timeframe to allow analysis of vocalizations and accompanying behaviors, as this species is documented to be sensitive to, and expressive of, changes in weather phenomena and daily cycle events [21]. The two focal animals were also observed in the two weeks following the eclipse to increase the sample for quantitative comparison of eclipse vs. non-eclipse vocalization bouts. 

### 2.3. Analysis

The behaviors of each focal species approaching, during, and immediately after totality were qualitatively compared to baseline behaviors for each taxon. Since the partial eclipse began more than an hour before and lasted more than an hour after totality, it was important to note behavioral changes as totality waxed and waned. However, the vast majority of the partial eclipses before and after totality resulted in nearly imperceptible light differences with only the few minutes (to the human eye, ~15 min) before and after totality becoming dramatically darker. Unfortunately, it is not possible to deduce when the animals might have become aware of the astrological phenomenon (if ever), and thus the notes included continuous annotation in an effort to perceive subtle behavioral changes. Ultimately, observed behaviors were ascribed to one or more of four overarching behavioral categories: normal (baseline), evening, apparent anxiety, and novel (Table 1 and Table 2). Examples of behaviors categorized as evening behaviors included roosting/nesting behavior, movement towards the entrances to evening dens, and increased activity in nocturnal animals. Examples of behaviors attributed to apparent anxiety were pacing, huddled group formations, and postural movements indicative of increased vigilance. Novel behaviors were those that fell outside of evening or apparent anxiety behaviors but which also seemingly deviated from the normal baseline behaviors.

In addition, an independent, quantitative analysis of siamang behavior while vocalizing was conducted using Raven Lite 2.0 (Bioacoustics Research Program, Cornell Lab of Ornithology). First, baseline recordings were analyzed in order to map normal bout structure (following [22]). The vocal repertoire was divided into organizing sequences (OSs) and great call sequences (GCSs), the former of which serves as preparation for the latter. Organizing sequences (OSs) were defined as beginning with a mutual, introductory call without a consistent structure, and terminating in a final “ululating scream”, a sound produced by the male in a single breath with fluctuating frequency [22]. The great call sequence in these siamangs is a highly organized with a consistent structure that begins with a series of short sequential barks produced by the male and ends with a series of short sequential barks produced by the female. Total bout duration was the time elapsed between the start of the first OS and the termination of the final GCS within each bout. For the purposes of analysis, the vocalization bouts recorded on the day of the eclipse were treated as a single sample (as the preliminary onset of the eclipse began shortly after commencement of the focal period), while the nine sequences recorded during the comparative calling bouts were analyzed both as nine individual samples of sequences and as a single combined sample of all sequences across these calling bouts. Two-tailed t-tests assuming unequal variance were then performed to compare OS and GSC durations on the day of the eclipse to both comparative samples.

## 3. Results

The observed behaviors of each species during the eclipse are presented in Table 2. Of the 17 species that were formally observed, 13 (76%) appeared to exhibit behaviors during the eclipse that differed from behavior observed outside of the eclipse (Table 2). In more than half of the taxa, a change in behavior was observed, including in most bird species and both (otherwise relatively sedentary) reptile species. Some of the change in behavior could be attributed to a response to the zoo’s visitors’ reactions to the eclipse (see limitations discussed below). However, that the behavior of many of the animals observed was similar to the behavior described in the literature during previous eclipses, the observed behavior changes may have been in direct response to the eclipse.

### 3.1. Baseline Behaviors

Three of the mammal species (grizzly bears, seals and sea lions) and one of the bird species (kookaburras) did not have noticeable behavioral changes leading up to or during the eclipse.

#### 3.1.1. Grizzly Bears

During baseline observations, the grizzly bears’ behavior consisted of moving throughout the exhibit, resting in the shade, and interacting with nearby exhibit objects and enrichment items. The bears occasionally interacted with each other, with one individual soliciting the majority of the interactions. In the early stages of the eclipse, the bears continued to rest. During totality, the bears awoke and moved to the side of the enclosure closest to the visitors’ window. Once totality ended, the bears resumed resting in the shade and interacting with enrichment items.

#### 3.1.2. Harbor Seals and Sea Lions

During baseline observations, both taxa’s behavior generally consisted of resting on the deck, swimming, interacting socially, and vocalizing. These animals showed high levels of activity, frequently exiting and reentering the water. When in the water, all observed individuals swam at the surface and sub-surface, following similar pathways through the enclosure with only occasional deviations. Social interactions among the sea lions consisted of tandem swimming, mouthing motions, and chasing one another. As feeding time approached, the animals frequently popped their heads out of the water close to the feeding deck and observed the feeding platform. On the day of the eclipse, the harbor seal appeared to look at zoo visitors more frequently than on the previous days—perhaps a reaction to the large, excited crowd. As the eclipse commenced, there were no major changes in the sea lions’ behavior, though the largest male sea lion was observed gently mouthing his left flipper, described by his keeper as a slight stereotypic behavior. During totality, this behavior increased in frequency. The other two larger male sea lions and the harbor seal swam, with the sea lions moving between the water and the deck and the seal remaining in the water throughout totality. During and after totality, the three smallest male sea lions sporadically sparred with each other and vocalized, with a fourth male joining occasionally. The single female of the group also occasionally participated in these interactions but primarily rested on the deck, intermittently vocalizing. After the peak of totality, the female and one of the smaller male’s social interactions increased. Following cessation of totality, baseline behaviors were quickly resumed by all animals.

#### 3.1.3. Kookaburras

Baseline behaviors consisted of sporadic flights between perch sites throughout the exhibit and the occasional use of “laugh” vocalizations. No discernible changes in behavior were observed prior to, during, or following the eclipse.

### 3.2. Evening Behaviors

The majority of species that appeared to have a behavioral change during the eclipse displayed behaviors typical of their evening routines. Specifically, gorillas, giraffe, elephants, lorikeets, frogmouths, cockatoos, lapwings and the Komodo dragon all performed behaviors that are all clearly or possibly indicative of their nighttime routines. Some (gorillas, giraffe, lorikeets and Komodo dragons) also displayed potential signs of anxiety (discussed below).

#### 3.2.1. Gorillas

The focal gorillas consisted of one adult male and three adult females. During baseline observations, the gorillas’ behavior mainly consisted of resting in the shade, performing social grooming, communally foraging throughout the exhibit, watching guests, and occasionally interacting with nearby structures (e.g., fence, exhibit glass, large rocks and enrichment items). One of the three females was observed to copulate with the male of the group. During the onset of the eclipse, no observable changes in behavior were recorded. However, during totality, the female gorillas approached the entrance to their indoor enclosure in their typical hierarchical order from the far side of their exhibit—a behavior repeated every evening prior to their being secured inside for the night. Immediately after totality, the gorilla group quickly resumed baseline behaviors.

#### 3.2.2. Giraffes

During baseline observations, the giraffes’ behavior mainly consisted of eating lettuce offered by the public, chewing cud, walking and foraging throughout the exhibit, and interacting with enrichment items. The giraffes occasionally interacted with one another by touching necks and/or grooming. The herd was often seen standing in the back of the exhibit, looking beyond the fence. Throughout the study, the two males regularly smelled the females. During the early stages of the eclipse, there was no observed change in behavior. At totality, the group ceased feeding and huddled together at the back of the exhibit, farther away from the guests and closer to the entrance of their barn. The group also displayed behaviors often associated with anxiety (see below). 

#### 3.2.3. Elephants

During baseline observations, the elephants’ behavior consisted of foraging activities throughout the enclosure, accompanied by ear flapping and tail swatting. The trunk was frequently used in the manipulation of nearby objects (typically sticks and/or hay), to touch other individuals, and to spray water. During the early stages of the eclipse, no observable changes in behavior were observed in either individual. Approaching totality, both elephants approached the entrance of their barn and following totality, the matriarch remained in the vicinity of the barn entrance, while the other female began roaming and foraging across the exhibit.

#### 3.2.4. Lorikeets

During baseline observations, both species of lorikeet (housed together) engaged in vocalizations at varying volumes, drinking nectar offered by visitors, flying between perches, resting in shaded areas, bathing, and communal preening. During these periods, the flock largely consisted of small groups dispersed throughout the exhibit, which largely behaved independently of one another. As totality approached, all lorikeets increased in activity level and in vocalization volume and frequency. At the peak of totality, the flock immediately flew en masse towards the end of their exhibit where their indoor enclosure and nest boxes are located. This was the only instance during observations that the flock behaved synchronously (something also observed in the flamingos). As totality ended, the flock fell silent and many individuals began preening. As time passed, the birds gradually increased their activity and vocalizations.

#### 3.2.5. Tawny Frogmouths

During baseline observations, the nocturnal tawny frogmouths spent much of their time “stumping,” perched in a posture resembling a tree branch, and exhibiting minimal movement except visual scanning of their surroundings in response to stimuli. Occasionally, they would make small postural movements, and quietly vocalize, particularly if approached by another bird. During totality, the birds fully opened their eyes, appeared highly vigilant, and increased their movement including preening, continually looking around the area, and moving on their perches—behaviors rarely seen during the baseline observations. After totality, the frogmouths resumed “stumping.”

#### 3.2.6. Cockatoos

During baseline observations, the male cockatoo spent most of his time inside nest boxes and rarely emerged, though he occasionally alternated between nest boxes. The female cockatoo largely occupied spaces outside of the nest box, and her pre-eclipse behavior consisted of preening and moving between perches. During totality, the male cockatoo emerged and interacted with the female: the pair touched beaks, preened one another, and occasionally raised their crests. After totality, the cockatoos ceased interacting and separated. Following totality, the male reentered a nest box.

#### 3.2.7. Lapwings

During baseline observations, the lapwings’ behavior consisted of foraging on the ground, preening, wing-flapping, and sporadic vocalization. Just prior to and during totality, the pair moved around the enclosure rapidly, vocalized loudly, and flapped their wings in a more exaggerated manner. After totality, the lapwings quickly resumed baseline behaviors.

#### 3.2.8. Komodo Dragon

During most of the baseline observation period, the solitary Komodo dragon was inactive, sitting motionless under a bush and only occasionally turning his head or flicking his tongue. As the eclipse commenced, there were no observable changes in behavior. Nearing totality, the dragon became momentarily active and altered his resting spot, then returned to inactivity. At totality, the individual rapidly approached the entrance to his evening den (which was closed, preventing him from entering). The Komodo dragon then ran erratically around his enclosure until the sun reappeared, at which time he returned to his prior state of near total inactivity.

### 3.3. Potential Anxiety Behaviors

After evening behaviors, the most common behaviors observed during the eclipse were those often associated with apparent anxiety. For instance, in addition to the evening behaviors performed by the gorillas, during the peak of the eclipse, the group’s male charged the glass. Likewise, while the giraffes huddled near the entrance of their evening barn during the eclipse, one male paced and began swaying his neck and body back and forth. As totality subsided, two males and one female galloped for several minutes, a behavior that was not witnessed during any other observation. Following totality, one of the same males paced for more than an hour through the end of the observation period. The remainder of the group returned to baseline behavior within approximately one hour after totality. Similarly, the movements of the Komodo dragon and the flocking behavior of the lorikeets (see above) appeared to some extent to replicate their evening behavior, but their frenetic nature also resembled potential anxiety. While these taxa exhibited potentially mixed responses to the eclipse, the baboons and flamingos seemed to display predominantly behaviors usually associated with apparent anxiety.

#### 3.3.1. Baboons

The Riverbanks Zoo’s baboon troop self-organizes into two typically separate patrilineal harems: one male-female pair and one sub-troop consisting of a dominant male with a harem of females. Prior to the eclipse, the troop’s behavior primarily consisted of moving throughout their exhibit while foraging, resting in the shade, interacting with enrichment items, and interindividual communications including allogrooming, lip smacking, and submissive displays almost exclusively by the subordinate male (m2) to the dominant male (m1) occasionally reinforced by threat yawns directed by m1 to m2.

In the preliminary stages of the eclipse, no discernible changes in behavior were observed. However, as totality approached, the two subgroups consolidated (notably huddling all together at several time points), appeared highly vigilant, and ran around the enclosure as a group. One female baboon repeatedly paced and walked in circles for approximately twenty-five minutes. As the eclipse waned, the group’s movements gradually slowed, accompanied with sporadic vocalizations by some individuals, and the original social groupings reformed. The atypical behaviors ceased within 30 min after totality. 

It should additionally be noted that, prior to totality, a helicopter passed over the enclosure and, almost simultaneously, a visitor group shouted. At this time, three of the baboons directed threat displays at the visitors. Thus, it is possible that the baboons may already have been experiencing elevated stress levels prior to the commencement of the eclipse; a circumstance which may have elevated the apparent anxiety of their response.

#### 3.3.2. Flamingos

Baseline behaviors among flamingos consisted of vocalizations of varying volume, movement in and out of ponds, foraging, resting in and out of shade, and social interactions (often characterized by disputes, which featured loud vocalizations and wing flapping). Throughout most of these observational periods, the flock remained divided into two or three subgroups. As totality approached, however, the flamingos decreased their interindividual distances and congregated tightly in the middle of their enclosure on a central island, with juveniles huddled in the center of the flock. The birds held their heads high, continuously looked around the area, vocalized loudly, and stood relatively motionless. This was the only instance of complete synchrony among the flock members during the observation periods. During totality, most adult flamingos ruffled their feathers and flapped their wings. As totality ended, some flamingos began lowering their heads and preening their feathers. The flock then gradually began to disperse more evenly throughout their enclosure, eventually occupying the same distribution that was typical during baseline observations. Even after totality, as the animals resumed their normal behavior, individuals seemed slightly vigilant with their heads up.

### 3.4. Relatively Novel Behaviors

The behavioral changes of two taxa (Galapagos tortoises and siamangs) were both notable and not easily ascribed to other categories, though, in both cases, they seemed rather frenetic and therefore possibly anxious during the eclipse.

#### 3.4.1. Galapagos Tortoises

During baseline observations, the Galapagos tortoises moved slowly around their enclosure, engaging in foraging behaviors or subtly interacting with other individuals. Between movements, they frequently rested in the shade or in a mud wallow. During interactions, the tortoises occasionally smelled one another, and both males grunted periodically when interacting with females. At the onset of the eclipse, no discernible changes in behavior were observed. However, immediately prior to totality the group huddled together and, just prior to totality, a pair of tortoises began mating. During totality, all four tortoises became more active, moved faster than had been seen during baseline observations and dispersed in various directions to different sections of the enclosure. Following totality, all tortoises gazed up at the sky, and then the group’s activity levels gradually decreased toward baseline levels.

#### 3.4.2. Siamangs

Prior to the eclipse, both the male and female siamang rested, groomed, and occasionally brachiated around their exhibit, expressing typical baseline behaviors. However, immediately prior to totality, both individuals began more vigorous brachiation and initiated loud vocalizations. The organizing sequences (OSs) to these bouts had no consistent pattern but began with a bark produced by the female and ended in an increase in the frequency of female barking, culminating in the final “ululating scream” produced by the male. The great call sequences (GCSs) began with a series of short sequential barks produced by the male and ended with a series of short sequential barks by the female. 

The total call duration during the eclipse was 388 seconds, significantly shorter than the average control duration of 673 seconds. The final GCS of the calling bout associated with the eclipse ended approximately 19 seconds prior to totality and lasted for 31.5 seconds, while during the control calls this portion averaged 39.9 seconds in length. Both the OS and GCS portions of the call during the eclipse were significantly shorter than those in the control samples (OS: x̅ = 45.0 vs. 74.4 s, respectively, *p* = 0.031, and GCS: x̅ = 32.7 vs. 39.9 s, respectively; *p* = 0.002, Table 3). Additionally, following the end of the final GCS of the eclipse call, both siamangs continued making disorganized vocalizations, whereas during control bouts, only the male made disorganized vocalizations following the final GCS.

## 4. Discussion

Approximately three-quarters of the focal species (13/17, 76%) observed during the eclipse exhibited behaviors that qualitatively differed from their behaviors before or after the eclipse. The behavioral anomalies were ascribed to one or more of three overarching behavioral categories: apparent anxiety, evening, and novel (Table 1). The most frequent response was an evening response, with animals mimicking their end-of-day or nighttime routines when exposed to the darkness of the eclipse. The taxa that engaged in evening behaviors were the gorillas, elephants, tawny frogmouths, cockatoos, lapwings, lorikeets, and Komodo dragon. The gorillas, elephants, and Komodo dragon all crossed their enclosure to approach the entrance to their nighttime quarters. All of the diurnal birds also displayed similar behaviors of returning to their evening roosts. In contrast, the nocturnal tawny frogmouth exhibited the opposite behavior, stirring from its roost to become active during the time of the eclipse. These behaviors indicate a standardized circadian disruption during which changes in light (and potentially temperature) appear to stimulate routine end-of-day behaviors.

Five of the seventeen focal taxa displayed behaviors in addition to, or in place of, evening routine behaviors that were interpreted by both researchers and zookeepers as apparently anxious. The most extreme expressions of apparent anxiety were observed in the giraffes, flamingos, and baboons. Throughout totality, giraffes exhibited several stereotypic behaviors indicative of anxiety, including swaying, group galloping, and the formation of tight huddles [23]. Indeed, one male individual continued to display anomalous behaviors for more than an hour after the eclipse ended. Both baboons and flamingos also clustered together during the eclipse, despite the propensity for both populations to normally divide into subgroups. Like the male giraffe described above, one female baboon also continued to pace for a substantial period of time following the eclipse—a behavior not previously observed during baseline observations or previously reported by the animal’s keeper. Interestingly, these seemingly anxious behaviors are contrary to previous observations of baboons observed in a zoo in Chile during an earlier eclipse event. During that eclipse, animals became less active, less aggressive, and increased grooming behaviors [7].

In addition to exhibiting evening behaviors, the gorillas and lorikeets also displayed muted anxiety responses. During the eclipse, the male gorilla charged the glass of the enclosure. Meanwhile, as totality approached, the lorikeets called more frequently and loudly before swooping en masse across their enclosure—a rare display of synchronous behavior that was otherwise not observed in this species.

Finally, the siamangs presented an unusual behavioral variant during the eclipse. Both the organizing sequences and great call sequences of their vocalizations were significantly shorter than they were outside of the eclipse. In addition, both the male and female calling bouts during the eclipse were disorganized after the end of the final GCS, while during non-eclipse days, only the male continued to vocalize following the final GCS. This cacophonous, shared post-call vocalization was similar to the beginning of an additional OS, but gradually stopped without initiation of a subsequent GCS. It is unclear whether this response should be interpreted as an anxious response, or simply an anomalous behavior induced by changes in light and/or atmospheric pressure, as siamangs have been documented to be sensitive to, and expressive of, changes in both weather phenomena and daily cycle events [21].

The behavior of the zoo animals during the eclipse was not that different than the previously documented behavior of their wild counterparts during eclipses. For example, the captive giraffes’ increase in activity culminating in communal galloping was also exhibited by wild giraffes in Zambia during a 2001 eclipse [24]. Likewise, the zoo’s zebras exhibited a muted response to the eclipse, which is similar to what was observed in wild zebras in Zimbabwe during a 2001 eclipse [18]. Another example is that many of the zoo’s birds engaged in evening behaviors, and using weather radars, Nilsson and colleagues [8] reported a broad decrease in bird biological activity, which implies that diurnal birds may have descended to roost.

Beyond the formally observed taxa, for three non-focal taxa (Malayan tapirs, zebras, and babirusas), accounts of animal behavior were relayed to the research team by zookeepers caring for these animals. While these animals were therefore not subjected to baseline behavioral observations, their keepers summarized the extent to which observed behaviors represented normal or anomalous behaviors for those particular individuals.

A female Malayan tapir awoke from sleep as the eclipse began and started foraging. As totality commenced, the tapir repeatedly entered and exited her pool until the eclipse ended, a behavior previously never seen by her keeper. Elsewhere, the zoo’s babirusas were observed to continue exploring their enclosure but became more alert and skittish during the eclipse. Finally, two zebras were observed to lay down together at the onset of totality, where they remained until cessation of the eclipse.

Despite the temptation to extrapolate our observations to the hypothesized reasons for captive species’ behavior, there are several caveats that must be emphasized. First and foremost, the impact of a consistent daily routine (and the disruption posed by the eclipse to this routine) upon the behavior of animals cannot be underestimated. Indeed, the most popular response to the eclipse was animals approaching their nighttime dens, roosts, or barns—a learned behavior derived from a consistently practiced evening routine that likely creates an association with the onset of darkness and a drop in temperature. One clear example of a changed routine was that the zoo’s seals and sea lions were fed an hour earlier than usual so that the feeding could take place before the onset of the eclipse. This change in routine is clearly a confounding variable that makes it impossible to distinguish the exact cause or causes of changes in behavior of the seal and sea lions.

Second, the influence of many more enthusiastic visitors than usual on the day of the eclipse on the behavior and stress levels of the animals cannot be discounted. Anthropogenic disturbances such as cheering, applause, and other background noises may have induced additional anxiety in the animals, as seen in the threat display of three baboons toward a visitor group shortly prior to the onset of the eclipse. While efforts were made to minimize the confounding anthropogenic effects on our data by conducting baseline behavioral observations during the zoo’s highest-capacity weekend, the increased human presence may have contributed to the behavioral responses of the animals (see reviews by [25,26]). One example is the grizzly bears waking during totality, which could be attributed to the meteorological phenomenon or to the cheering crowd during the event. Furthermore, two other external stimuli may have obfuscated behavioral interpretation: fireworks near the zoo during totality and a helicopter flying overhead during the eclipse. As secondary sources of intrusive noise (e.g., construction work) have been shown to induce stress and/or anxiety in captive animals [27], both events—in addition to the aforementioned crowd disturbances—may have further heightened the animals’ anxiety and affected their behavior. However, given that most species exhibited behaviors associated with their evening routines—behaviors that should not be triggered by anthropogenic exuberance—at least some of the observed behavioral differences seem to be related to the eclipse. In addition, the apparent anxiety-related behaviors correspond to other published observations of wild animals during eclipses that likely were much less stimulated by the presence of people.

Another limitation of this study is that it is mostly qualitative instead of quantitative. However, the literature on animal behavior during eclipses is scarce and conflicting, and so we opted to study more taxa for a short period of time rather than conduct longer observations on fewer species. Furthermore, the eclipse occurred on one of the busiest days in the history of the zoo, and on one of the hottest days of the summer, and we worried that including a broader time sample may have caused its own confounding variables by bracketing a wider range of temperatures and weather conditions, and by risking greater variance in visitor density between the baseline and focal periods of observation. Thus, to eliminate the possibility of more confounding variables that could affect animal behavior even without an eclipse, a mostly qualitative study was conducted. We believe that the discussion of the behaviors of many animals during this rare event adds to the paucity of knowledge on how animals behave during eclipses.

Future observational studies of a similar nature may wish to account for several variables in a more controlled manner—especially increased anthropogenic presence. One approach might be to count the number of visitors at each enclosure to enable comparisons of heavily versus lightly visited enclosures, or to close several exhibits to the public to provide an isolated response of certain animals to an eclipse without anthropogenic interference. Observing predatory species during eclipses may provide insight into variations in response mode and/or magnitude as a result of trophic level. Indeed, it could be hypothesized that, as prey species typically express greater vigilance and possess a stronger fight-flight-or-freeze response mechanism, prey may be more susceptible to changes in environment than top-level carnivores.

Although documenting the responses of wild animals to total solar eclipses is logistically more complex than observing captive animals (especially in terms of taxonomic breadth observable by individual research teams), such data would provide valuable information for understanding and interpreting behavioral responses to anomalous external stimuli relative to these observations of zoo animals. To this end, several crowdsourced efforts such as the *Life Responds* project, a Citizen Science initiative that encouraged members of the public to document wildlife during the solar eclipse, may prove a valuable resource by multiplying the geographic breadth of recorded data. While efforts to standardize data collection methodologies are necessary, similar future initiatives may be able to quickly add a wealth of information on the responses of animals to eclipses. Indeed, by 2030, future solar eclipses will transect the wild habitats of several species in this study, including baboons, siamangs, elephants, giraffes and flamingos in addition to countless taxa not analyzed herein.

Collectively, this and future research into animal behaviors during rare phenomena have the potential to expand our understanding of the reactions of a diversity of animals to novel stimuli, and the extent to which response modes and magnitudes vary between taxa exposed to the same stimulus. While efforts are necessary to more closely control the environment to minimize obfuscation by anthropogenic stimuli, repeating studies on these and novel taxa—both of which may include a more quantitative component of behavioral categorization—will continue to enhance our knowledge of the complex interplay between environmental stimuli and behavioral responses across vertebrates.

## 5. Conclusions

The high proportion of taxa (76%) observed to exhibit a behavioral response to the eclipse suggests a strong potential impact of this meteorological phenomenon on modifying animal activity. That the predominant response mimicked evening/nighttime behaviors suggests that the change in light (and possibly temperature) associated with the eclipse served as a major cue in behavioral determination. However, five taxa exhibited behavioral responses associated with apparent anxiety, including swaying and huddling. While several potential anthropogenic factors may have contributed to this response, our finding that these behaviors aligned with previously published observations of wild animal responses during the eclipse suggest a causal relationship between the eclipse and these responses. Nonetheless, we conclude that future observational studies of this nature may wish to minimize the obfuscating potential of this pressure by closing certain exhibits to the public to isolate the impact of certain animals to the eclipse without anthropogenic interference. 

## Figures and Tables

**Table 1 animals-10-00587-t001:** Ethogram defining potential behavioral responses to the eclipse, and their associated classification.

Behaviors	Broad Categorization	Description of Behavior
Resting	Normal/baseline	Animal remains at rest, either sleeping or consciously inactive
Grooming	Normal/baseline	Animal participates in grooming behaviors, either of itself (e.g., preening) or of other group members
Foraging	Normal/baseline	Animal explores enclosure to discover food
Playing (alone or gregariously)	Normal/baseline	Animal is actively engaged in play or similar dynamic activity, either alone or with other group members
Approach evening enclosure	Evening	Animal approaches its evening den, barn, or enclosure
Bird song (evening repertoire)	Evening	Animal engages in its dusk or evening vocalizations
Group consolidation	Apparent Anxiety	Animals form cluster by huddling together
Coordinated running	Apparent Anxiety	Animals engage in synchronized group running
Pacing	Apparent Anxiety	Animal engages in short-term, repetitive locomotor patterns
Swaying	Apparent Anxiety	Animal remains stationary but moves body from side to side
Increased vigilance/vocalization	Apparent Anxiety	Animals increase volume and/or frequency of vocal calls; animals scan surroundings for threat and protect vulnerable parties
Aggressive behaviors	Apparent Anxiety	Animal engages in conflict or performs combative displays; animal charges window of enclosure
Atypical vocalizations	Novel	Animal engages in previously unobserved calling bouts of an unusual mode and duration
Skyward gazing	Novel	Animal looks repeatedly towards the sky

**Table 2 animals-10-00587-t002:** Focal taxa observed during the 2017 total solar eclipse. Behavioral observations among focal taxa during the total solar eclipse. Behaviors are classified into one of four overarching categories: evening/nighttime response, apparently anxious response, novel response, or no response (e.g., behaviors indistinguishable from baseline behaviors).

Order	Species	Sample (M:F)	Observed Behaviors during Eclipse	Behavioral Interpretation
Mammals	Western lowland gorilla (*Gorilla gorilla gorilla*)	1:3	Approached den enclosure entrance; aggressive behaviors by group male	Evening (all individuals); possible anxiety (male)
	Hamadryas baboon (*Papio hamadryas*)	2:4	Group consolidation; coordinated running (all individuals); pacing (one individual)	Apparent anxiety (all individuals)
	Siamang gibbon (*Symphalangus syndactylus*)	1:1	Increased movement; vocalization with atypical structure and end	Novel response (unusual behavioral variant)
	Giraffe (*Giraffa cf. camelopardalis*)	2:5	Group consolidation; approached barn enclosure; pacing; coordinated running; swaying	Evening; apparent anxiety
	African elephant (*Loxodonta africana*)	0:2	Approached barn enclosure	Evening
	Grizzly bear (*Ursus arctos horribilis*)	2:0	Awoke from sleep; minor movement	Baseline
	California sea lion (*Zalophus californianus*)	4:1	Swimming; vocalizations; mouthing flipper (one individual)	Baseline
	Harbor seal (*Phoca vitulina*)	1:0	Swimming; vocalizations	Baseline
Birds	American flamingo (*Phoenicopterus ruber*)	31:16	Group consolidation around juveniles; increased vigilance and vocalizations	Apparent anxiety
	Rainbow and coconut lorikeets (*Trichoglossus moluccanus* and *Trichoglossus haematodus*)	17:26	Coordinated swooping; group consolidation around nest boxes; increased vocalizations followed by communal silence	Evening; apparent anxiety
	Tawny frogmouth (*Podargus strigoides*)	1:1	Increased movement and activity	Evening
	Major Mitchell’s cockatoo (*Lophochroa leadbeteri*)	1:1	Increased activity; shared interactions	Evening
	Masked lapwing (*Vanellus miles*)	1:1	Increased activity	Evening
	Laughing kookaburra (*Dacelo novaeguineae*)	1:1	Occasional movement and vocalizations	Baseline
Reptiles	Galapagos tortoise (*Chelonoidis nigra*)	2:2	Increased movement (group consolidation followed by scatter); shared interaction; gazing skyward	Novel response
	Komodo dragon (*Varanus komodoensis*)	1:0	Increased movement; approached den enclosure	Evening; possible anxiety

**Table 3 animals-10-00587-t003:** Quantitative observations of the durations of organizing and great call sequences performed by siamangs during the day of the eclipse and during nine control bouts. *p*-values are derived from a two-tailed *t*-test between eclipse bouts and control bouts.

Measurement Variable	Organizing Sequences (OSs, in Seconds)		Great Call Sequences (GCSs, in Seconds)	
	Eclipse	Control	Eclipse	Control
Average duration (standard deviation)	45.0 (±12.5)	74.4 (±41.7)	32.7 (±5.1)	39.9 (±10.2)
Minimum and maximum sequence lengths	34.7–63.8	4.4–173.7	25.3–39.4	22.2–65.3
*p*-value	0.031		0.002	

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
