# Peer review of "Total Eclipse of the Zoo: Animal Behavior during a Total Solar Eclipse"

_animals, 2020, doi:10.3390/ani10040587_

Round 1
Reviewer 1 Report
This manuscript is well written and data presented are reflective of a unique event. Data are interesting as a reflection of the unique phenomena they observed but have limited significance beyond that.
My main concerns/confusion on the manuscript focused on the methods/results and are described below.
- Line 134: Manuscript describes how three species were “monitored” but in a different and less systematic way than the other species. This seems to distract/pull away from the larger study. This summary could be kept as an appendix but seems different enough to not be comparable and included in the main body.
- Line 151: “… notes on each animal’s status were made at thirty-minute intervals.” What does this mean? How was it defined, utilized and where is it interpreted in the results?
- Line 159: “The behaviors of each focal species approaching, during, and immediately after totality were qualitatively compared to baseline behaviors…” Is baseline behavior not that same as approaching? Or is baseline the two days before and approaching is minutes before? More explanation of how data were collected within these intervals and what defined these intervals are needed to understand methods.
- Line 160: “… were qualitatively compared” Why were behaviors qualitatively compared when systematic observations including continuous and scan sampling were used? Location data were also recorded? It seems unclear why these systematic methodologies were described only to then rely on qualitative comparisons. In the introduction it seemed implied that previous evaluations of this phenomena were limited to qualitative assessments and thus required a quantitative assessment. If this manuscript relies solely on qualitative comparisons how is it different from previous evaluations? I think it would be of significant value to this manuscript to conduct a more quantitative comparison. The qualitative summaries are nice, and could be included as an appendix for reference, but it seems if this line of inquiry needs to occur the analysis should be reflective of modern quantitative assessments. Even basic quantitative comparisons of how behavior varied would be informative. More “advanced” analysis such as comparisons of behavioral diversity between conditions or something along those lines would boost the analysis of this manuscript. The siamang call data is an excellent example of how quantitative analysis provides meaningful context and descriptions of behavior around this phenomena in ways that have not previously been done.
Author Response
Reviewer 1:
This manuscript is well written and data presented are reflective of a unique event.
Thank you!
Data are interesting as a reflection of the unique phenomena they observed but have limited significance beyond that.
Indeed, we agree that these data are really most valuable relative to this extremely rare phenomenon. However, in preparation for this research, we learned 1) that there was very little previous research on the subject, and 2) that there was enormous interest in these questions from both our scientific colleagues and especially from the general public. We hope that this study can especially inform future investigations during subsequent solar eclipses – research that is clearly not about fundamental behavioral questions, but rather is fascinating almost because of its rarity.
My main concerns/confusion on the manuscript focused on the methods/results and are described below.
Line 134: Manuscript describes how three species were “monitored” but in a different and less systematic way than the other species. This seems to distract/pull away from the larger study. This summary could be kept as an appendix but seems different enough to not be comparable and included in the main body.
We agree, and have moved this section to the discussion.
Line 151: “… notes on each animal’s status were made at thirty-minute intervals.” What does this mean? How was it defined, utilized and where is it interpreted in the results?
We updated the text to clarify.
Line 159: “The behaviors of each focal species approaching, during, and immediately after totality were qualitatively compared to baseline behaviors…” Is baseline behavior not that same as approaching? Or is baseline the two days before and approaching is minutes before? More explanation of how data were collected within these intervals and what defined these intervals are needed to understand methods.
We added much more detail about this. In short, the partial eclipse bracketing totality lasted more than two hours (more than an hour before and after). However, it was only in the few minutes (to the human eye ~15minutes before and after) that the sky was noticeably darker. Of course we cannot be sure when (or if) the animals were aware of the phenomenon.
Line 160: “… were qualitatively compared” Why were behaviors qualitatively compared when systematic observations including continuous and scan sampling were used? Location data were also recorded? It seems unclear why these systematic methodologies were described only to then rely on qualitative comparisons. In the introduction it seemed implied that previous evaluations of this phenomena were limited to qualitative assessments and thus required a quantitative assessment. If this manuscript relies solely on qualitative comparisons how is it different from previous evaluations? I think it would be of significant value to this manuscript to conduct a more quantitative comparison. The qualitative summaries are nice, and could be included as an appendix for reference, but it seems if this line of inquiry needs to occur the analysis should be reflective of modern quantitative assessments. Even basic quantitative comparisons of how behavior varied would be informative. More “advanced” analysis such as comparisons of behavioral diversity between conditions or something along those lines would boost the analysis of this manuscript. The siamang call data is an excellent example of how quantitative analysis provides meaningful context and descriptions of behavior around this phenomena in ways that have not previously been done.
We COMPLETELY agree that doing a quantitative comparison would be much more scientifically interesting. However, because of 1) the short nature of the phenomenon, 2) the small sample sizes of most of the species that we were observing, and 3) our desire to observe as many focal taxa as possible (since there had previously been so little literature on the subject on which to base a priori hypotheses), there was no way to collect as much data as we would have needed to conduct a proper quantitative analysis on any but the siamangs. (And, frankly, we were lucky that the siamangs proved interesting enough for quantitative analysis. The other taxon that we targeted for video data recording – the lorikeets – did not prove to have quantifiable reaction other than the broad flocking behavior.) Taken on their own, 1) because the eclipse was so short, we could not gather enough data points on any individual animals to quantify differences in behavior. At best, we might have been able to say how rare (or unique) those behaviors were, relative to baseline, but upon completing that work, it turned out to be more speculative and seemed rather qualitative anyway. Likewise, 2) for almost all focal taxa, the sample sizes were too small to even quantify the percent of individuals behaving in specific ways. Again, with sample sizes that small, even the quantitative analyses would be skewed by the qualitative personalities of individual animals. The only way to have truly quantifiable behavior would have been to watch individual animals for much more time (e.g., instead of spreading the 356 hours of notes over more than a dozen exhibits, we could have focused on individual taxa). However, we were not sure which animals would provide the most interesting observations and didn’t want to miss results – thus 3) we chose to spread ourselves at that rate – and we were very concerned about the other confounding factors that would have been introduced if we extended our observation durations. Namely a) the eclipse took place on one of the hottest days of the year and if we had watched the animals for many days before or after, then we could not determine how the variation in the weather was confounding our results. More importantly, the eclipse was likely the busiest day ever at the zoo. Thankfully the two days preceding it (Saturday and Sunday the 19th and 20th of August 2017) were also exceptionally busy (as the phenomenon drew many people to Columbia in anticipation of the eclipse). However, if we observed the animals even a few days prior or subsequent, then the anthropogenic factor would have been completely different. For instance, there were probably 1/10th as many visitors to the zoo on the day after the eclipse than there were during the eclipse and thus the animals might have behaved differently. Obviously it would have been best to observe the animals with no people at all, both during the eclipse and for “baseline”, but, as that was not possible, it was important to try to match crowd size as best as possible. Based on our results, we can now recommend certain taxa to observe during future eclipses more quantitatively and in more ideal circumstances. However, without our data, future scientists would be as ‘stuck’ as we were in making predictions. For instance, we can now recommend that our colleagues try to quantify interesting behaviors in siamangs, giraffe, etc. in more ideal circumstances to confirm observations, but would not recommend that they establish an extensive observation of elephants, grizzly bears or pinnipeds as these species did not seem to exhibit very strong reactions and are therefore maybe not worth the investment, especially if other more probable taxa can be observed instead. That is, in our opinion, the real value of our unfortunately mostly qualitative study. We think our colleagues will appreciate this work in preparation for future eclipses and hope the journal agrees!
Reviewer 2 Report
This manuscript is very well written and structured. Regarding the species descriptions of behavioral responses to the eclipse, I would recommend dropping the baseline responses from the manuscript. You wouldn’t lose much from the manuscript but this change would help cue the reader that each description presented is one that differed from baseline. You described behavioral changes for some species (bears, seals) and I found myself referring back to your table to see how you had characterized it. I also wonder if structuring the descriptions around the type of response vs the taxa might be more meaningful, as the reader can than compare similarities in responses across taxa. I also have questions regarding the description of your observational methods. You noted that continuous and scan sampling methodologies were used. However, given the focus on qualitative descriptions of behavior, it appears ad libitum sampling is the most appropriate description of these methods. It appears from your descriptions that you were recording anything of note which would most closely describe ad lib sampling. If you recorded everything a single animal was doing during observations (e.g. durations of behavior) that would qualify as continuous sampling, but that doesn’t appear to be what occurred. Scan sampling would imply data were recorded at pre-determined intervals on a group of animals and you do state you took notes on each animals status at 30 min intervals, but from that description I’m not sure that qualifies scan sampling. If there were more standardized methods that were employed for selecting individuals and recording behavior you should include those, otherwise this does appear to be ad lib sampling. That said, I appreciate your defense of your methods in the discussion and can understand their purpose for the question you were studying. Overall the writing was very strong. I did note on like 460 that there should be a comma between “that” and “as” and then either delete “than” or include that it was comparing to predators.
Author Response
Reviewer 2:
This manuscript is very well written and structured.
Thank you!
Regarding the species descriptions of behavioral responses to the eclipse, I would recommend dropping the baseline responses from the manuscript. You wouldn’t lose much from the manuscript but this change would help cue the reader that each description presented is one that differed from baseline. You described behavioral changes for some species (bears, seals) and I found myself referring back to your table to see how you had characterized it.
Thank you for this suggestion. We have revised the manuscript to make this more clear.
I also wonder if structuring the descriptions around the type of response vs the taxa might be more meaningful, as the reader can than compare similarities in responses across taxa.
This is perhaps the single most valuable piece of feedback that we have received and we completely agree that it improves the manuscript.
I also have questions regarding the description of your observational methods. You noted that continuous and scan sampling methodologies were used. However, given the focus on qualitative descriptions of behavior, it appears ad libitum sampling is the most appropriate description of these methods. It appears from your descriptions that you were recording anything of note which would most closely describe ad lib sampling.
Indeed this is what we did and we have made this adjustment.
If you recorded everything a single animal was doing during observations (e.g. durations of behavior) that would qualify as continuous sampling, but that doesn’t appear to be what occurred. Scan sampling would imply data were recorded at pre-determined intervals on a group of animals and you do state you took notes on each animals status at 30 min intervals, but from that description I’m not sure that qualifies scan sampling. If there were more standardized methods that were employed for selecting individuals and recording behavior you should include those, otherwise this does appear to be ad lib sampling.
We added text to clarify this. In short, we did ad libitum focal observation on each individual in each group for all but the two largest groups (the two large bird flocks) for which overall group and sub-group behavior were described. Although the objective was to have continuous notes on each animal for the smaller groups, out of fear that some individuals might be missed (especially during more behaviorally chaotic periods or during the excitement of the eclipse), we made sure that each research team noted all group positions and behaviors every 30-minutes to supplement the hopefully complete focal observations. In other words, this was a backup to catch missed behaviors. In the end, these snapshots proved useful qualitative summaries, but we don’t think that the research teams ended up missing individual behaviors even at chaotic times. All notebooks are available for review.
That said, I appreciate your defense of your methods in the discussion and can understand their purpose for the question you were studying. Overall the writing was very strong.
Thank you so much. This feedback much improves the manuscript!
I did note on like 460 that there should be a comma between “that” and “as” and then either delete “than” or include that it was comparing to predators.
Done!